# Differential Associations of PIVKA-II with Epithelial and Mesenchymal Features in HCC and PDAC

**DOI:** 10.3390/ijms26157581

**Published:** 2025-08-05

**Authors:** Antonella Farina, Gaia Cicolani, Valentina Viggiani, Matteo Maini, Antonio Angeloni, Emanuela Anastasi

**Affiliations:** Department of Experimental Medicine, Sapienza University of Rome, Policlinico Umberto I, 00181 Rome, Italy; antoeffe22@gmail.com (A.F.); gaia.cicolani@uniroma1.it (G.C.); valentina.viggiani@uniroma1.it (V.V.); mn.matteo@gmail.com (M.M.); antonio.angeloni@uniroma1.it (A.A.)

**Keywords:** HCC, PDAC, PIVKA-II, EMT

## Abstract

Hepatocellular carcinoma (HCC) and pancreatic ductal adenocarcinoma (PDAC) are aggressive malignancies characterized by a poor prognosis and resistance to conventional therapies. Mounting evidence suggests the pivotal role of epithelial–mesenchymal transition (EMT) in tumor progression, metastasis, and therapeutic resistance in these cancers. Protein induced by vitamin K absence II (PIVKA-II)—a valuable HCC detector—has ultimately emerged as a potentially relevant biomarker in PDAC, serving as both a serum biomarker and a prognostic indicator. This study investigates the putative link between PIVKA-II expression and the EMT process in HCC and PDAC. Using a Western blot analysis and electrochemiluminescence immunoassay (ECLIA), we quantified PIVKA-II serum levels alongside two canonical EMT markers—Vimentin and E-cadherin—in selected cohorts. Emerging data suggest a dual, context-dependent role for PIVKA-II. Beyond its diagnostic value in both malignancies, its co-expression with EMT markers points to a potential mechanistic involvement in tumor invasiveness and phenotypic plasticity. Notably, the selective detection of E-cadherin in HCC implies limited EMT activation and a preservation of the epithelial phenotype, whereas the higher expression of Vimentin in PDAC reflects a more substantial shift toward EMT. We provide a comprehensive analysis of key molecular markers, their involvement in EMT-driven pathophysiological mechanisms, and their potential as novel diagnostic tools.

## 1. Introduction

Hepatocellular carcinoma (HCC) and pancreatic ductal adenocarcinoma (PDAC) are among the most aggressive solid tumors, both characterized by late-stage diagnosis, a poor prognosis, and marked resistance to conventional therapies [1,2]. PDAC is generally considered to be more aggressive than HCC since it is often diagnosed at an advanced stage due to the lack of effective early detection methods. It tends to invade surrounding tissues (including nerves and blood vessels) early on, and metastases frequently occur at an early stage, particularly in the lymph nodes and distant organs [3]. Otherwise, HCC shows a more heterogeneous clinical course, although its prognosis is often better than that of PDAC, particularly when diagnosed early through surveillance in patients with chronic liver disease or cirrhosis. Compared to PDAC, HCC often remains confined to the liver and, although it can metastasize, this typically occurs at a later stage of the disease.

Over the past decade, scientific research has provided substantial evidence supporting the central role of epithelial–mesenchymal transition (EMT)-induced transcriptional reprogramming in carcinogenesis. This process contributes to the generation of heterogeneous tumor cell populations, including hybrid subpopulations displaying intermediate features between epithelial and mesenchymal phenotypes [4,5,6,7]. The transition from an epithelial to a mesenchymal phenotype is well established and thoroughly documented to be marked by a series of molecular alterations, including reduced expression of E-cadherin, aberrant regulation of E-cadherin/β-catenin complexes, and increased expression of Vimentin [8,9]. During EMT, the loss of E-cadherin expression—a marker of epithelial integrity—is associated with the upregulation of Vimentin, a mesenchymal cytoskeletal protein linked to enhanced migratory capacity. This shift in expression reflects the acquisition of mesenchymal traits as well as correlating with increased tumor aggressiveness, invasive potential, and poor clinical outcomes [8,9]. In light of these insights, the close interplay between EMT-induced cellular transformations and tumor progression emerges as a critical focal point in the diagnostic landscape of HCC and PDAC. The relationship between Vitamin K and EMT is an emerging and potentially valuable avenue of investigation within the field of cancer biology. Among the few contributions to this novel field, researchers demonstrated that vitamin K3 (menadione) is capable of suppressing EMT in human colorectal cancer cells [10,11].

To date, no clinically validated biomarkers are available to specifically monitor EMT-driven processes in HCC and PDAC [12,13]. This diagnostic gap highlights the urgent need for novel biomarkers capable of capturing EMT activity, which could greatly enhance early detection, risk stratification, and therapeutic guidance in these aggressive tumors. Among the emerging biomarkers gaining increasing attention in HCC and PDAC research, protein induced by vitamin K absence II (PIVKA-II) appears to be one of the most promising diagnostic tools [14]. PIVKA-II, also known as des-γ-carboxy prothrombin (DCP), is an abnormal prothrombin precursor resulting from defective γ-carboxylation, a vitamin-K-dependent post-translational modification occurring in hepatocytes [15]. Originally developed and validated as a biomarker for the diagnosis and monitoring of HCC, PIVKA-II has more recently been implicated in other gastrointestinal malignancies, including PDAC [16,17,18,19]. Beyond its clinical application as a serum biomarker, emerging data suggest a broader biological role for PIVKA-II in tumorigenesis. Given the distinct levels of biological aggressiveness, metastatic potential, and tumor microenvironmental context of HCC and PDAC, it is plausible that PIVKA-II may exert tumor-type-specific effects on EMT regulation. These differences could reflect diverse molecular pathways through which PIVKA-II contributes to tumor progression, invasiveness, and phenotypic plasticity in each cancer.

The present study thus aims to investigate the potential role of PIVKA-II at the intersection of EMT and tumor progression in HCC and PDAC, in order to better understand its dual function as a biomarker and a putative driver of cancer pathogenesis.

## 2. Results

To better assess the reliability of PIVKA-II as a biomarker, circulating levels of this protein were evaluated in a cohort comprising 30 patients with HCC and 37 with PDAC, all of whom were newly diagnosed at the time of enrollment. Previous studies have suggested that a PIVKA-II threshold of 30 ng/mL may serve as a feasible and shared diagnostic cut-off for both HCC and PDAC [20]. The clinical characteristics of HCC and PDAC patients enrolled for this study are summarized in Table 1.

Among the 30 patients diagnosed with HCC, 25 showed PIVKA-II levels above the established threshold, yielding a positivity rate of 83%, indicative of high diagnostic sensitivity. Similarly, 32 out of 37 patients with PDAC exceeded the same threshold, resulting in an 86% positivity rate and further supporting the biomarker’s strong diagnostic potential. To further evaluate the performance of conventional prognostic biomarkers, AFP (alpha-fetoprotein) and CA19.9 (carbohydrate antigen 19-9) levels were measured using the COBAS e801 immunoassay system. In the HCC cohort, 37% of patients (11/30) were AFP-positive, whereas 83% were PIVKA-II-positive (*p* = 0.0086). In the PDAC cohort, 92% (34/37) were CA19.9-positive, and 86% were PIVKA-II-positive (*p* = 0.0225). These findings support the potential utility of PIVKA-II as a robust diagnostic and prognostic biomarker in both tumor types (Table 2).

### 2.1. Detection of PIVKA-II and EMT Markers in the Sera of Patients with HCC and PDAC by Western Blot Analysis

To investigate the protein expression of PIVKA-II, Vimentin, and E-cadherin, we performed a Western blot analysis on serum samples from patients with HCC and PDAC. A total of twelve serum samples from patients with HCC and PDAC, along with one positive control derived from pooled sera known to be positive for PIVKA-II, E-cadherin, and Vimentin, were analyzed. As is shown in Figure 1, a band corresponding to PIVKA-II was detected at the expected molecular weight of 67 kDa in samples from both patients with HCC and PDAC, confirming its molecular presence. A band at 57 kDa, consistent with Vimentin, was also observed in samples from patients with both tumor types, although with variable intensity among samples. In contrast, a 130 kDa band corresponding to E-cadherin was exclusively detected in samples from patients with HCC, highlighting a differential expression pattern of EMT markers between HCC and PDAC.

Western blot analysis of human serum samples, performed as described in Section 4. In the upper panel, lane 1 contains the molecular weight marker (M); lanes 2–5 contain sera from patients with PDAC; and lanes 6–9 contain sera from patients with HCC. Lane 10 (CTRL) represents a pooled serum sample used as a positive control. Coomassie Blue staining of the membrane (lower panel) is shown as a loading control; the prominent band corresponds to serum albumin. A representative experiment out of three independent replicates is shown. Quantification of serum albumin was performed using ImageJ (version 1.54p), and the results represent the mean of three independent replicates.

### 2.2. Vimentin and E-Cadherin Thresholds

In a further comparative analysis, serum levels of the EMT markers Vimentin and E-cadherin were evaluated in patients with HCC and PDAC. To assess their diagnostic potential in differentiating between these two malignancies, receiver operating characteristic (ROC) curves were generated for both markers. For Vimentin, a previously established ROC analysis [18] identified a cut-off value of 487 ng/mL, with an area under the curve (AUC) of 74%, sensitivity of 73%, and specificity of 72% (95% CI). In contrast, the ROC curve for E-cadherin revealed a cut-off value of 6.4 ng/mL, associated with an AUC of 87%, sensitivity of 80%, and specificity of 81% (95% CI) (Figure 2).

### 2.3. Comparative Analysis of Circulating PIVKA-II, Vimentin, and E-Cadherin in Human Sera

Based on the established cut-off values for EMT markers, we found that 77% of patients with HCC (23/30) were E-cadherin-positive, while only 10% (3/30) were Vimentin-positive. In contrast, among patients with PDAC, 46% (17/37) were E-cadherin-positive and 81% (30/37) were Vimentin-positive (Figure 3A,B).

## 3. Discussion

HCC and PDAC both remain among the most lethal gastrointestinal cancers, often being diagnosed at advanced stages and being notoriously resistant to standard therapeutic approaches. Although AFP and CA19.9 are currently recommended as reference biomarkers for HCC and PDAC, respectively, several studies have highlighted significant limitations in their specificity, restricting their utility in early diagnosis and accurate disease stratification [12,13]. Therefore, the identification of robust biomarkers, which can not only aid in diagnosis but also reflect the biological behavior of tumors, remains a critical unmet clinical need in the era of personalized medicine. In the present study, we highlight the multifaceted role of PIVKA-II, a vitamin-K-dependent abnormal prothrombin, classically associated with HCC diagnosis, now emerging as a relevant biomarker in PDAC. The diagnostic potential of PIVKA-II has been confirmed by the results presented here with the detection of serum levels above the diagnostic threshold in 83% of patients with HCC and 86% of patients with PDAC, suggesting a high sensitivity of this biomarker in both pathological contexts. In the case of HCC, the clinical use of PIVKA-II has already been validated by several studies and has been included in the diagnostic and surveillance guidelines of the Japanese Society of Hepatology (JSH) [21,22,23,24]. Due to its association with adverse clinical features, like vascular invasion (e.g., portal vein thrombosis), high tumor proliferation, and a poor prognosis, it is used in combination with other biomarkers, such as AFP and AFP-L3, to enhance diagnostic accuracy [25,26,27]. In contrast, the role of PIVKA-II in the context of PDAC, remains less defined. Our study contributes to this by strengthening the hypothesis of its potential relevance in malignancy. Although the relationship is not fully understood, previous data are compelling since it has been demonstrated that PIVKA-II may serve as a more promising biomarker than currently available markers, such as CA19.9, offering greater diagnostic accuracy [8,9]. This evidence is supported by in vitro data showing that PIVKA-II is overexpressed in PDAC tissues. This may reflect underlying alterations in vitamin-K-dependent pathways during pancreatic tumorigenesis. Furthermore, the fact that previous observations have reported that Panc-1 cells, a widely used PDAC model, are capable of producing PIVKA-II, reinforces the plausibility of its tumor-specific expression [26]. Taken together, these findings suggest that PIVKA-II may reflect the presence of disease and provide insight into the molecular derangements that are characteristic of pancreatic cancer. Further investigation into its clinical utility and biological significance is warranted. The novelty of our work lies in investigating the relationship between PIVKA-II and EMT in HCC and PDAC. Our combined immunometric assay and Western blot analyses confirmed the presence of PIVKA-II in sera from patients with both cancer types and uncovered differential co-expression patterns with canonical EMT markers: Vimentin and E-cadherin [28,29]. Vimentin was detected with both cancer types, but with a higher prevalence in patients with PDAC. The strong association between PIVKA-II and Vimentin observed in patients with PDAC suggests that PIVKA-II plays a functional role in modulating cellular signaling pathways involved in invasiveness, migration, and phenotypic plasticity. In this context, PIVKA-II could either promote EMT or represent an epiphenomenon of its activation. Conversely, E-cadherin was predominantly observed in patients with HCC, suggesting a less aggressive epithelial phenotype. It is noteworthy that PIVKA-II is produced in conditions of vitamin K deficiency or in the presence of vitamin K antagonists, due to the incomplete γ-carboxylation of prothrombin [10,30]. In order to ascribe a potential role for PIVKA-II in EMT, several considerations must be taken into account. A vitamin K deficiency has been documented in patients with PDAC, primarily due to exocrine pancreatic insufficiency, which impairs the absorption of essential nutrients, including fat-soluble vitamins such as vitamin K [25]. This condition may result in a functional vitamin K deficiency in patients with PDAC, contributing to the aberrant γ-carboxylation of prothrombin and the consequent production of PIVKA-II [26]. Furthermore, studies have demonstrated that PIVKA-II production in PDAC cell lines, including Panc-1, is modulated by glucose availability, highlighting a potential interplay between cellular metabolism and vitamin K status in the tumor microenvironment [31]. These findings all underscore the clinical and pathophysiological significance of vitamin K deficiency in patients with PDAC, with important implications for the use of PIVKA-II as a tumor biomarker. Some preclinical studies have shown that vitamin K2 (menaquinone) can exert anti-proliferative and anti-invasive effects on tumor cell lines, potentially counteracting EMT [10,27,28,32]. This suggests that PIVKA-II may not only reflect a functional vitamin K deficiency but also play an active role in the activation of pro-metastatic programs in specific tumor contexts. Our results reinforce and confirm those of previous in vitro studies, since in vitro studies, conducted on HCC cell lines, have revealed that the loss or dysfunction of E-cadherin can increase tumor aggressiveness by promoting cell migration and invasion. Epigenetic activation of E-cadherin is a candidate therapeutic target in human hepatocellular carcinoma [33,34]. The association between PIVKA-II and E-cadherin in HCC suggests that, in this context, the protein is more strongly expressed during early stages of tumor progression, when the epithelial phenotype is still preserved. This may reflect a role for PIVKA-II more closely related to proliferation, rather than migration or metastatic dissemination. This observation is consistent with the literature data suggesting that PIVKA-II-positivity does not necessarily correspond to advanced EMT, but may be a sign of specific alterations in neoplastic hepatocytes in response to vitamin K deficiency [20,30].

## 4. Materials and Methods

### 4.1. Patients

This retrospective study enrolled patients who were referred to the Tumor Markers Laboratory of Policlinico Umberto I, Rome, Italy, for the evaluation of serum EMT markers.

Serum samples were selected from two diagnostic categories/populations:Thirty patients affected by HCC;Thirty-seven patients affected by PDAC.

Patients met the following inclusion criteria: newly diagnosed at the time of enrollment, no prior treatment with neoadjuvant therapy, no serious disabilities, and the absence of diabetes. Patients were excluded in cases of higher than a single serving a day in terms of alcohol consumption, if they had evidence of active hepatopathy, if they were taking vitamin K antagonists, or had any coagulopathy. A detailed medical history was collected for each subject upon enrollment. Peripheral blood samples were collected and analyzed for PIVKA-II, Vimentin, and E-cadherin levels. Demographic variables, including sex distribution, were comparable between the two patient groups, with a predominance of Caucasian participants. All subjects were over 18 years of age and provided written informed consent prior to inclusion in the study. The study protocol was approved by the Institutional Review Board of Policlinico Umberto I (Prot. Number 208/13) and conducted in accordance with the Declaration of Helsinki.

### 4.2. Blood Collection and Storage

Blood samples were collected in yellow-top Vacutainer tubes, clotted for 60–90 min, and then centrifuged at 3000× *g* rpm for 10 min according to a standard protocol [18]. Serum was aliquoted and stored at −80 °C until analysis.

### 4.3. Western Blot Analysis

Selected human sera were diluted 1:320 in 1X PBS and Laemmli Sample Buffer as previously reported [20]. A total of 10 μL of diluted serum was loaded onto a 7.5% and 12% SDS-PAGE system, along with a molecular weight marker (PiNK Plus, GeneDirex, Las Vegas, NV, USA or Opti-Protein XL Marker, Abm, Richmond, BC, V6V 2J5, Canada), according to the manufacturer’s instructions. Separated proteins were then transferred to nitrocellulose membranes (Bio-Rad, Hercules, CA, USA), as described elsewhere [19]. Membranes were first stained with Red Ponceau and then probed with the following primary mouse antibodies: anti PIVKA-II (1:1000, Byorbit, Explore, Bioreagents, Cambridge, UK); anti-Vimentin (1:200, Santa Cruz Biotechnology, Dallas, TX, USA); and anti-E-cadherin (1:200; Santa Cruz Biotechnology, Dallas, TX, USA). As a secondary antibody, polyclonal anti-mouse IgG-HRP (1:10,000) was used. Detection was performed using Western Bright (Advansta, Menio Park, CA, USA) following the manufacturer’s instruction [31]. Coomassie staining was performed following the manufacturer instruction (Coomassie brilliant Blue R-250 BIO-RAD, Hercules, CA, USA).

### 4.4. Immunometry

Serum concentrations of PIVKA-II were measured in ng/mL using the Elecsys^®^ PIVKA-II immunoassay on the Roche COBAS e411 platform (Roche Diagnostics, Basel, Switzerland), which utilizes a one-step sandwich electrochemiluminescence immunoassay (ECLIA) methodology, as previously described [20]. The analytical range of the assay was 3.5–12,000 ng/mL, with a lower limit of detection (LoD) of ≤3.5 ng/mL and a lower limit of quantification (LoQ) of ≤4.5 ng/mL [35]. All measurements were performed in duplicate, strictly adhering to the manufacturer’s instructions. Serum levels of Vimentin and E-cadherin were quantified using manual two-step sandwich ELISA assays. Vimentin was measured using the Human Vimentin (VIM) ELISA Kit (CUSABIO, Houston, TX, USA) and E-cadherin using the Human E-cadherin ELISA Kit (Elabscience Bionovation Inc., Houston, TX, USA), both according to the respective manufacturer’s instructions. All assays were carried out in duplicate.

### 4.5. Statistical Analysis

Descriptive statistics were used to summarize demographic and clinical data, expressed as means or frequencies (percentages). The diagnostic accuracies of PIVKA-II, Vimentin, and E-cadherin were assessed using receiver operating characteristic (ROC) curve analysis. Performance metrics for PIVKA-II, Vimentin, and E-cadherin were reported as the area under the curve (AUC). A 95% confidence interval (95% CI) for the AUC was also estimated. Comparisons between the HCC and PDAC groups regarding marker positivity were conducted using the unpaired *t*-test. The distribution of Vimentin and E-cadherin positivity across different PIVKA-II concentration ranges in PIVKA-II-positive individuals was assessed using the chi-square test. A two-tailed *p*-value of < 0.05 was considered to be statistically significant. All statistical analyses were performed using MedCalc Statistical Software, version 14.8.1.

### 4.6. Densitometric Analysis

Quantification of serum albumin bands was performed by densitometric analysis using the Image J software (1.47 version, NIH, Bethesda, MD, USA), which was downloaded from the NIH website (http://imagej.nih.gov, accessed on 1 August 2022).

## 5. Conclusions

Finally, our findings suggest that PIVKA-II may act as a ‘cross-road’ molecule, linking diagnostic utility to biological function, particularly in the context of EMT. On the one hand, its usefulness as a serum biomarker for diagnosing both HCC and PDAC is confirmed; on the other hand, its association with EMT markers, particularly in pancreatic cancer, suggests its potential involvement in tumor aggressiveness and invasiveness mechanisms. From a clinical perspective, dual detection of PIVKA-II and EMT markers could provide additional prognostic information and help to stratify patients based on tumor aggressiveness and EMT status. Furthermore, differential expression of EMT markers between patients with HCC and PDAC underlines the importance of interpreting biomarkers in the context of the disease, which could inform the development of tailored therapeutic strategies. In conclusion, future studies are warranted to delineate the functional interplay between PIVKA-II and EMT machinery, ideally through mechanistic investigations and larger clinical cohorts, to validate its role as a prognostic and therapeutic biomarker in gastrointestinal oncology. A limitation of our study is the absence of functional assays (such as gene knockdown or overexpression experiments) to validate the biological role of PIVKA-II in EMT-related processes. Nonetheless, we emphasize that our study was designed from a clinicopathological perspective, based on patient-derived tissue analyses. This “bedside-to-bench” approach aims to identify relevant expression patterns that may serve as a foundation for future mechanistic and translational research. Future prospective studies with long-term follow-up will be necessary to assess the prognostic significance of the findings of this study.

## Figures and Tables

**Figure 1 ijms-26-07581-f001:**
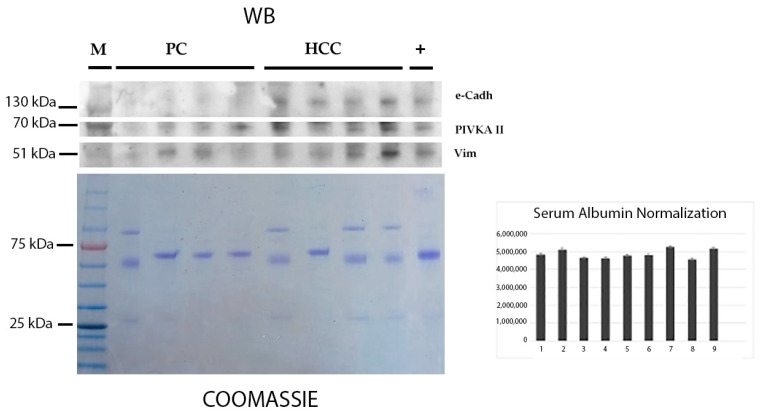
PIVKA-II, E-cadherin, and Vimentin in sera from patients with HCC and PDAC in WB.

**Figure 2 ijms-26-07581-f002:**
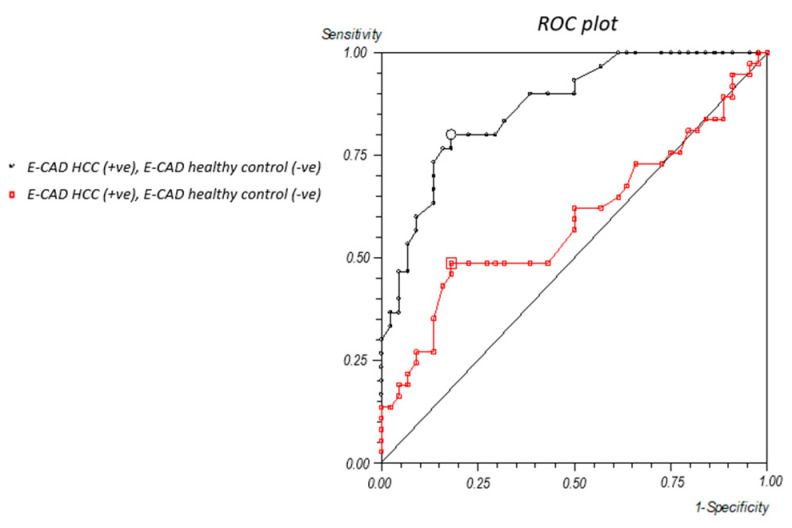
Receiver operating characteristic (ROC) curve for E-cadherin. Suggested threshold is 6.4 ng/mL.

**Figure 3 ijms-26-07581-f003:**
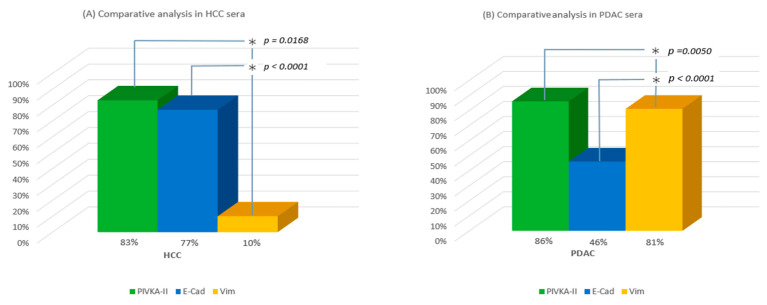
Comparative analysis of PIVKAII vs. Vimentin vs. e-cadherin in human sera of patients with HCC (**A**) and PDAC (**B**).

**Table 1 ijms-26-07581-t001:** Demographics and clinical characteristics of HCC and PDAC study cohort.

Parameters	Patients with HCC	Patients with PDAC
**Age**	73 ^1^	68 ^1^
	(54–85)	(41–89)
**Sex**		
Male	19/30 male	20/37 male
Female	11/30 female	17/30 female
**Grade**		
I–II	18/30 (60%)	16/37 (43%)
III–IV	12/30 (40%)	21/37 (57%)
**Stage**		
I–II	16/30 (53%)	10/37 (27%)
III–IV	14/30 (47%)	27/37 (73%)

^1^ Years (median and range).

**Table 2 ijms-26-07581-t002:** Comparison of PIVKA-II presence vs. AFP and CA19.9 in patients with HCC and PDAC, respectively.

Sera	PIVKA-II	AFP	CA19.9
**HCC**			
**Number of patients**	83% (25/30)	37% (11/30)	-
**PDAC**			
**Number of patients**	86% (32/37)	-	92% (34/37)

## Data Availability

The data presented in this study are available on request from the corresponding author.

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
