# Peer review of "Differential Associations of PIVKA-II with Epithelial and Mesenchymal Features in HCC and PDAC"

_ijms, 2025, doi:10.3390/ijms26157581_

Round 1
Reviewer 1 Report
Comments and Suggestions for Authors
1、It is recommended to add discussion on the association between Vitamin K and the EMT pathway in the Introduction section.
2、Research on PIVKA-II concerning the EMT pathway is limited; it is suggested to add experiments investigating the direct regulatory role of PIVKA-II knockout on the EMT pathway.
3、The discussion regarding E-cadherin in HCC contains excessive speculation; it is recommended to supplement with relevant in vitro experiments for validation.
4、It is recommended to supplement the ethics approval number for the sample source.
5、The authors are advised to ensure consistent formatting throughout the manuscript paragraphs.
Author Response
Rew1
- It is recommended to add discussion on the association between Vitamin K and the EMT pathway in the Introduction section.
Thank you for the recommendation, we have added the following paragraph in the revised version of the manuscript:
“The relationship between Vitamin K and EMT is an emerging and potentially valuable avenue of investigation within the field of cancer biology. Among the few contributions to this novel field, researchers demonstrated that vitamin K3 (menadione) is capable of suppressing EMT in human colorectal cancer cells
- Research on PIVKA-II concerning the EMT pathway is limited; it is suggested to add experiments investigating the direct regulatory role of PIVKA-II knockout on the EMT pathway.
We thank the reviewer for this valuable suggestion. We agree that investigating the direct regulatory role of PIVKA-II on the EMT pathway—such as through knockout experiments—would offer important mechanistic insights. However, due to current technical and time constraints (the time it takes to generate a knockout pattern can vary greatly, but generally takes several months) we are not able to include such experiments at this stage. Our study focus on a clinicopathological approach, starting from patient-derived observations rather than a purely molecular hypothesis. In this sense, our work represents a different and complementary starting point—from bedside to bench—aimed at identifying biologically and clinically relevant patterns that can subsequently guide mechanistic studies. We believe this patient-centered direction offers a solid rationale for further investigation of PIVKA-II in the context of EMT, which we plan to pursue in future work.
To better clarify this weakness we added the following comment in the conclusion:
“A limitation of our study is the absence of in-depth functional and molecular analyses; however, our approach is primarily clinicopathological, grounded in patient-derived observations rather than a hypothesis driven purely by molecular data. In this context, our work offers a distinct and complementary perspective—moving from bedside to bench—with the goal of identifying biologically and clinically relevant patterns that may inform future mechanistic investigations”.
- The discussion regarding E-cadherin in HCC contains excessive speculation; it is recommended to supplement with relevant in vitro experiments for validation.
We thank the reviewer for the insightful comment and fully acknowledge the importance of in vitro studies in elucidating the molecular mechanisms regulating E-cadherin expression. In response, we have cited two additional studies that demonstrate the expression and modulation of E-cadherin in HCC models and we have added the following paragraph:
“Our results reinforce and confirm previous in vitro studies since in vitro studies, con-ducted on HCC cell lines, have revealed that the loss or dysfunction of E-cadherin can in-crease tumor aggressiveness by promoting cell migration and invasion. Epigenetic activation of E-cadherin is a candidate therapeutic target in human hepatocellular carcinoma”.
Nevertheless, we believe that one of the major strengths of our study lies in the in vivo identification of these alterations within the physiopathological context of HCC. Our findings are based on clinical specimens and therefore reflect biologically and clinically relevant phenomena occurring in human tumor tissues. While in vitro validation could certainly complement these results, our primary objective was to provide robust in vivo evidence, which we consider a significant and translationally meaningful contribution to the current literature.
- The authors are advised to ensure consistent formatting throughout the manuscript paragraphs.
Ok, done.

Reviewer 2 Report
Comments and Suggestions for Authors
Through the use of Western blotting and electrochemiluminescence immunoassay (ECLIA), the authors assessed serum levels of PIVKA-II alongside two established epithelial-mesenchymal transition (EMT) markers, Vimentin and E-Cadherin, in specific cohorts of PDAC and HCC patients. The study demonstrated the selective presence of E-cadherin in hepatocellular carcinoma (HCC), indicating minimal EMT activation and a retention of the epithelial phenotype, while elevated levels of Vimentin in pancreatic ductal adenocarcinoma (PDAC) signify a more pronounced transition towards EMT. The authors presented a detailed examination of significant molecular markers, their roles in EMT-related pathophysiological processes, and their potential utility as innovative diagnostic tools. This is a very brief note that needs further exploration using bioinformatic tools and a more complete discussion of serum-based biomarkers in the literature for these cancers.
Comments
- Please provide histology or information about the tumor stage, grade of the patient cohorts, and if this is related to the levels of the biomarkers studied.
- Were there correlations between age and any of the biomarkers, as this might be a confounding variable?
- The manuscript will benefit from a bioinformatic-led confirmation from publicly available databases that allow for prognostic impacts of gene expression in larger cohorts of patients: https://kmplot.com/analysis/; https://www.cbioportal.org/
- Ponceau S staining of the membrane is difficult to see. Was there any quantification performed for serum albumin and averaged across the 3 replicates? Were the levels of the markers normalized by serum albumin? Please label the sizes of the control markers for Ponceau S.
- The authors stated that markers “were 100 CA19.9-positive, and 86% were PIVKA-II-positive (p = 0.0225)”. This finding is only borderline significant due to the low patient numbers; however, it appears more robust for HCC.
- What were the ROC curves for AFP (alpha-fetoprotein) and CA19.9 (carbohydrate antigen 19-9)?
- In the discussion, the authors need to elaborate on the utility of CA19.9 see (Azizian, A., Rühlmann, F., Krause, T. et al. CA19-9 for detecting recurrence of pancreatic cancer. Sci Rep 10, 1332 (2020). https://doi.org/10.1038/s41598-020-57930-x; Ballehaninna UK, Chamberlain RS. The clinical utility of serum CA 19-9 in the diagnosis, prognosis and management of pancreatic adenocarcinoma: An evidence based appraisal. J Gastrointest Oncol. 2012 Jun;3(2):105-19. doi: 10.3978/j.issn.2078-6891.2011.021. PMID: 22811878; PMCID: PMC3397644; Brian Haab, Lu Qian, Ben Staal, Maneesh Jain, Johannes Fahrmann, Christine Worthington, Denise Prosser, Liudmila Velokokhatnaya, Camden Lopez, Runlong Tang, Mark W. Hurd, Gopalakrishnan Natarajan, Sushil Kumar, Lynette Smith, Sam Hanash, Surinder K. Batra, Anirban Maitra, Anna Lokshin, Ying Huang, Randall E. Brand, A rigorous multi-laboratory study of known PDAC biomarkers identifies increased sensitivity and specificity over CA19-9 alone,Cancer Letters, https://doi.org/10.1016/j.canlet.2024.217245.) and suggest how PIVKA-II is an improvement.
Author Response
Rew 2
Through the use of Western blotting and electrochemiluminescence immunoassay (ECLIA), the authors assessed serum levels of PIVKA-II alongside two established epithelial-mesenchymal transition (EMT) markers, Vimentin and E-Cadherin, in specific cohorts of PDAC and HCC patients. The study demonstrated the selective presence of E-cadherin in hepatocellular carcinoma (HCC), indicating minimal EMT activation and a retention of the epithelial phenotype, while elevated levels of Vimentin in pancreatic ductal adenocarcinoma (PDAC) signify a more pronounced transition towards EMT. The authors presented a detailed examination of significant molecular markers, their roles in EMT-related pathophysiological processes, and their potential utility as innovative diagnostic tools. This is a very brief note that needs further exploration using bioinformatic tools and a more complete discussion of serum-based biomarkers in the literature for these cancers.
Comments
- Please provide histology or information about the tumor stage, grade of the patient cohorts, and if this is related to the levels of the biomarkers studied.
We thank the reviewer for this insightful comment. Histological classification, tumor stage, and grade information for the patient cohorts have now been included in the revised manuscript. Please, refer to Table 1. We have also performed a preliminary correlation analysis between tumor stage/grade and the expression levels of the biomarkers investigated in this study. While a clear trend was observed, statistical significance was limited by the sample size and cohort heterogeneity.
- Were there correlations between age and any of the biomarkers, as this might be a confounding variable?
We thank the reviewer for raising this important point. To address this concern, we performed correlation analyses between patient age and the expression levels of the biomarkers investigated in our study. No significant associations were observed between age and PIVKA-II, E-cadherin, or vimentin, suggesting that age is unlikely to represent a major confounding factor in our cohort. This lack of correlation is consistent with previous studies reporting that the expression of E-cadherin and vimentin is not significantly influenced by patient age in various tumor types (PMC9535065, EXCLI J. 2020;19:1225–1235). Similarly, the expression of PIVKA-II has been shown to be independent of age in hepatocellular carcinoma cohorts (PMC9364642).
- The manuscript will benefit from a bioinformatic-led confirmation from publicly available databases that allow for prognostic impacts of gene expression in larger cohorts of patients: https://kmplot.com/analysis/; https://www.cbioportal.org/
We thank the reviewer for this valuable suggestion. We fully acknowledge the relevance of bioinformatic validation using publicly available databases such as KMplot and cBioPortal. However, due to current time and resource constraints, we are unable to perform these analyses at this stage. We nonetheless consider this an important future direction and plan to address it in follow-up studies.
- Ponceau S staining of the membrane is difficult to see. Was there any quantification performed for serum albumin and averaged across the 3 replicates? Were the levels of the markers normalized by serum albumin? Please label the sizes of the control markers for Ponceau S.
We appreciate this detailed comment. We have replaced the Ponceau staining with the Blue Coomassie staining, and we have also added the quantification performed for serum albumin, averaged across the three replicates. However, the levels of the markers were not normalized to serum albumin, as the marker bands do not correspond in size to the albumin band. As suggested, we have now specified the molecular weight of the markers in the Figure 1.
- The authors stated that markers “were 100 CA19.9-positive, and 86% were PIVKA-II-positive (p = 0.0225)”. This finding is only borderline significant due to the low patient numbers; however, it appears more robust for HCC.
We thank the reviewer for this insightful comment. We agree that the statistical significance observed in the PDAC cohort (p = 0.0225) should be interpreted cautiously given the limited sample size. However, we would like to emphasize that CA19.9, while widely used, suffers from well-known limitations in specificity, as its levels can be elevated in various benign and inflammatory conditions (e.g., cholangitis, pancreatitis, biliary obstruction). In this context, the comparable positivity rate of PIVKA-II (86%)—despite the borderline statistical significance—suggests that it may represent a valuable complementary biomarker with potentially greater specificity. This is especially relevant in differentiating malignant from non-malignant pancreatic conditions. The key focus of our study is to investigate whether PIVKA-II may be involved in the early events of epithelial–mesenchymal transition (EMT), with the aim of identifying a robust marker of the early stages of pancreatic carcinoma. Unlike CA19-9, which is not an early-stage marker, PIVKA-II could potentially provide valuable diagnostic information at a stage when intervention is more likely to be effective.
- What were the ROC curves for AFP (alpha-fetoprotein) and CA19.9 (carbohydrate antigen 19-9)?
As the sensitivity and specificity of AFP and CA19.9 for HCC and PDAC, respectively, have already been extensively evaluated in previous studies using ROC curve analysis, we chose not to repeat this analysis in the present work (PMID: 27227515, PMID: 31223261 )
- In the discussion, the authors need to elaborate on the utility of CA19.9 see (Azizian, A., Rühlmann, F., Krause, T. et al. CA19-9 for detecting recurrence of pancreatic cancer. Sci Rep 10, 1332 (2020). https://doi.org/10.1038/s41598-020-57930-x; Ballehaninna UK, Chamberlain RS. The clinical utility of serum CA 19-9 in the diagnosis, prognosis and management of pancreatic adenocarcinoma: An evidence based appraisal. J Gastrointest Oncol. 2012 Jun;3(2):105-19. doi: 10.3978/j.issn.2078-6891.2011.021. PMID: 22811878; PMCID: PMC3397644).
Thank you for pointing this out. We added this sentence:
“While CA19.9 remains the standard biomarker for PDAC, its limited specificity—especially in the presence of obstructive jaundice or inflammation—reduces its diagnostic reliability (21-22). Moreover, recent studies highlight the need for more specific biomarkers or combined panels to improve diagnostic accuracy (23). In this context, PIVKA-II may offer added value, particularly given its performance in our PDAC cohort and its lower likelihood of being influenced by benign conditions (23)”.

Round 2
Reviewer 1 Report
Comments and Suggestions for Authors
1. The term “dual role” in the title is not substantiated by the data, as only differential associations were observed. The authors are advised to revise the title to more accurately reflect the study’s descriptive nature.
2. The statement in the limitations that “in-depth mechanistic studies are lacking” is too vague. Please add specific points, such as the absence of functional assays (e.g., gene knockdown/over-expression) and the lack of prospective survival data.
3. Reference 19 contains two DOI numbers. Please verify and correct the citation so that only the correct and unique DOI is retained.

Author Response
We sincerely thank the reviewer for their valuable comments and suggestions, which helped us to improve the quality of the manuscript.
- The term “dual role” in the title is not substantiated by the data, as only differential associations were observed. The authors are advised to revise the title to more accurately reflect the study’s descriptive nature.
We thank the reviewer for this insightful comment. We agree that the term “dual role” may imply a mechanistic dichotomy that is not fully supported by our data, which primarily show differential associations. To address this concern, we have revised the title to better reflect the descriptive and observational nature of our study.
The new title is: “Differential associations of PIVKA-II with epithelial and mesenchymal features in HCC and PDAC.
- 2. The statement in the limitations that “in-depth mechanistic studies are lacking” is too vague. Please add specific points, such as the absence of functional assays (e.g., gene knockdown/over-expression) and the lack of prospective survival data.
We thank the reviewer for this important observation. We have revised the limitations section to include more specific points. The revised sentence is “A limitation of our study is the absence of functional assays (such as gene knockdown or overexpression experiments) to validate the biological role of PIVKA-II in EMT-related processes. Nonetheless, we emphasize that our study was designed from a clinicopathological perspective, based on patient-derived tissue analyses. This "bedside-to-bench" approach aims to identify relevant expression patterns that may serve as a foundation for future mechanistic and translational research. Future prospective studies with long-term follow-up will be necessary to assess the prognostic significance of the findings”
- OK the DOI references has been corrected
Reviewer 2 Report
Comments and Suggestions for Authors
No further comments
Author Response
We sincerely thank the reviewer for their valuable comments and suggestions, which helped us to improve the quality of the manuscript.